

# BorFIT: A Novel LiDAR-Based Training Dataset for Individual Tree Segmentation and Species Detection in northern boreal Forests

Jacob Schladebach[1], Birgit Heim[1], Léa Enguehard[1], Mareike Wieczorek[1], Jakob Broers[1], Robert Jackisch[2], Josias Gloy[1], Kunyan Hao[1], James Tretton[1], Anna Gorshunova[1], and Stefan Kruse[1]

[1]Alfred Wegener Institute Helmholtz Centre for Polar and Marine Research, Potsdam, Germany
[2]Technical University of Berlin, Berlin, Germany

**Correspondence:** Jacob Schladebach (jacob.schladebach@awi.de) and Stefan Kruse (stefan.kruse@awi.de)

**Abstract.** BorFIT is a novel training data set designed to assist in the segmentation of individual trees and the detection of species from LiDAR point clouds, thus contributing to deep learning-based forestry applications. Recent advancements in AI-supported individual tree detection have shown significant progress; however, satisfactory results remain elusive in dense and structurally-complex boreal forests. We compiled a training data set designed to remedy this issue. It comprises 384 LiDAR point clouds, each with an area of 20 m × 20 m, in the form of reference plots, with up to 200 manually segmented and species classified trees per point cloud. We carried out LiDAR surveys at 146 sites between 2021 and 2024 in East Siberia (Yakutia), northwest Canada, and Alaska (USA), selected along a bioclimatic gradient to represent the circumboreal region. From each LiDAR transect derived point cloud, we extracted a minimum of four reference plots (each 20 m × 20 m) based on maximum tree heights within the plots to systematically sample the apparent tree density gradient. We manually segmented identifiable trees within each reference plot point cloud leading to 16,530 individual trees in total. Following segmentation, we trained four randomForest classifiers to predict the species of every segmented tree. The predicted tree species include: *Picea mariana* (Britton, Sterns Poggenb.), *Picea sitchensis* ((Bong.) Carrière), *Picea glauca* ((Moench) Voss), *Pinus contorta* (Douglas ex Loudon), *Abies lasiocarpa* ((Hook.) Nutt.), *Larix laricina* ((Du Roi) K.Koch), *Betula papyrifera* (Marshall), *Betula neoalaskana* ((Regel) Ashburner McAll.), *Populus balsamifera* (L.), *Populus tremuloides* (Michx.), *Pinus sylvestris* (Thunb.) and *Alnus glutinosa* ((L.). The data offer the means for 3D space analysis of species distribution and stand structure around the circumboreal region. Furthermore, it can be used as a training data set for artificial intelligence (AI) applications and thereby improve our understanding of the boreal forest's vegetation reorganization in response to significant global warming.





# 1 Introduction

The boreal forest, or taiga, occupies approximately 25% of the world's forested land and plays a pivotal role in climate change
mitigation (Thiffault, 2019; Kauppi, 2014). This northernmost forest biome is located between the tundra and temperate decid-
uous forests, accounting for about 12% of global forest biomass (Pan et al., 2013). The high latitudes, and therefore the boreal
region, are warming twice as fast as the global average, which leads to significant changes in vegetation structure (Scheffer
et al., 2012). These changes have resulted in alterations to the distribution of biomass in boreal forests extending beyond the
Arctic treeline, with an advancement of broadleaf tree species into previously conifer-dominated stands (Myneni et al., 1997).
Monitoring its development under significant global warming is crucial due to its ecological attributions, e.g. carbon seques-
tration. Large-scale monitoring at the individual tree level is necessary to evaluate these structural changes while delivering
additional information regarding forest health and responses to environmental changes. The most common approaches for tree
segmentation rely on 2D true colour or infrared colour images and area-based techniques like watershed segmentation (Chehreh
et al., 2023). Conventional area-based methods relying on canopy height models tend to overlook trees especially when dealing
with thin and overlapping crowns coupled with a dense understory layer, as demonstrated by Brieger et al. (2019). Uncrewed
aerial vehicle (UAV)-based LiDAR (Light Detection and Ranging) point clouds present a cost-effective method to monitor large
areas. LiDAR sensors obtain detailed 3D point clouds of objects like trees and shrubs using laser pulses sent by the sensor. The
pulses are reflected by a given object and the return-signal is recorded by the sensor. By measuring the Time-of-Flight (ToF),
the distance to the object is calculated. This allows to differentiate between twigs, branches, vegetation and ground. When
properly segmented, using these point clouds one can quantify forest structure and biomass volume (Popescu, 2007; Xu et al.,
2021). Recent advancements in artificial intelligence (AI) supported tree detection within LiDAR point clouds have shown
promise; however, achieving satisfactory results specifically in boreal forests remains a challenge. Current segmentation tech-
niques based on AI highly depend on the selection of appropriate algorithms for the local conditions, while challenges in dense
forests with high biodiversity remain (Falkowski et al., 2008; Chehreh et al., 2023; Hastings et al., 2020; Ma et al., 2022). The
segmentation of trees is a prerequisite for object-based analyses like species assignment. The performance of the assignment
algorithm is therefore highly dependent on the quality of the training data set (Chehreh et al., 2023). High-resolution data sets
covering a diverse range of species and environmental conditions from the boreal region are scarce. van Geffen et al. (2022)
generated a data set to provide training data for AI applications, and focused on structure from motion (SfM) point clouds of
forest plots, which include canopy height models and orthoimages. While this data set offers valuable insights into the struc-
ture of Siberian boreal forests, it does not provide detailed assessments at the individual tree level. The LiPheStream data set
by Wittke et al. (2024) includes point clouds of 458 individual boreal trees from Finland and offers high temporal resolution
through an 18-month monitoring phase. However, this individual tree data set is less representative of the broader boreal region
and is more suited for analyzing individual tree development. Here we present the novel training data set BorFIT with manually
segmented trees that were assigned a species using a randomForest classifier. It can act as a training data set for AI appications.
Furthermore, large-scale forest structure analysis based on global canopy height models (CHM) (Lang et al., 2023; Potapov
et al., 2021) (GEDI) could be assessed through data sets like BorFIT. BorFIT comprises 384 LiDAR point clouds collected



from 146 locations across bio-climatic gradients in Siberia, Canada, and Alaska, with each point cloud containing up to 200 individual trees with assigned species. The tree species classificaiton was performed using a randomForest classifier based on a training data set employing both structural (LiDAR) and RGB information.

## 2 Data and methods

The methods used to obtain the final data are described in the following sections. The UAV surveys conducted during expeditions yielded raw point cloud data (n = 146) which went through several processing steps including manual tree segmentation and species classification (ntree = 16530) (Fig. 1).

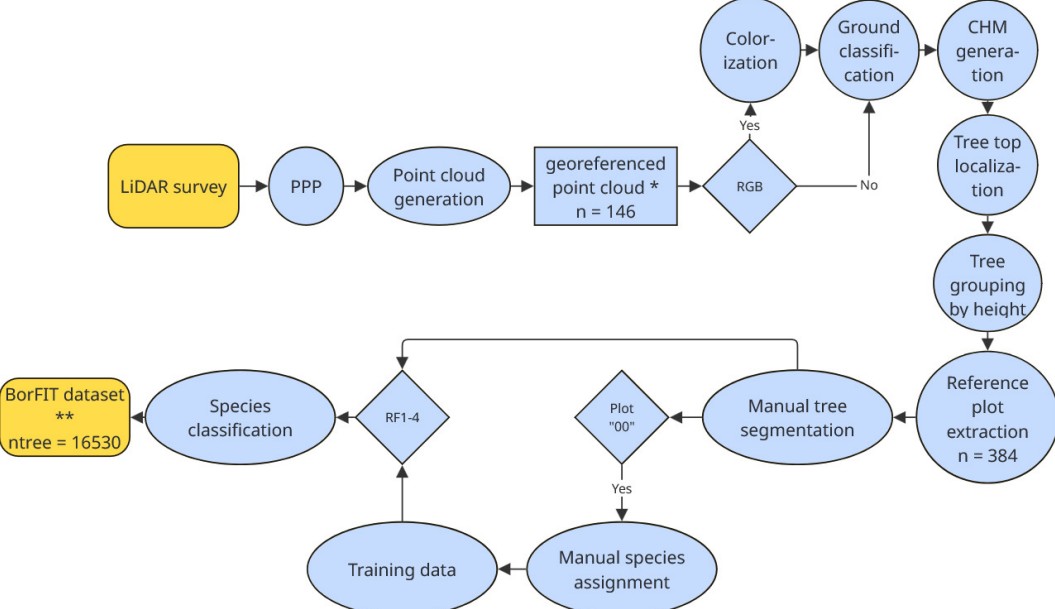

**Figure 1.** Flow chart of processing steps from UAV survey to BorFIT data set. Abbreviations: Precise point positioning (PPP), Canopy height model (CHM), Red green blue (RGB), randomForest models 1-4 (RF1-4), * (Kruse et al., 2025e, c, a, b), ** (Kruse et al., 2025f)



Earth System
Science
Data

## 2.1 Study region

Since 2021, we carried out UAV LiDAR surveys in addition to extensive multisensor drone overflights in various locations within the boreal region of Eastern Siberia (Morgenstern et al., 2023), Canada, and Alaska (Fig.2, Table 1). In each region, we established forest transects across several thousands of kilometers covering both the tundra-taiga-ecotone where the forest transitions into tundra grasslands and the dense mixed forests further to the south. In Northwest Canada and Alaska, the latitudinal treeline represents the northernmost extent of the research area. In Northwest Canada the transect reaches from the broadleaf forest in British Columbia up to the Mackenzie Delta where sparse forests dissolve into the Tundra. The Alaskan transect begins at the southern coastal region, reaching northward into interior Alaska up to the southern slopes of the Brooks range. The 2024 expedition covers the Seward Peninsula in western Alaska. In eastern Siberia (2021) the transect is located between the Lena and Aldan river interfluve in the central Yakutian lowland up to the Verkhoyansk mountain region. We established the transects with the objective of enabling long-term monitoring of ecosystem dynamics and responses to climate change and disturbances such as wildfires. Accordingly, we positioned sites prior to fieldwork guided by satellite derived change detection and forest density as well as optical features such as fire scars and products of time since last fires using f.e. the Hansen landcover product (Hansen et al., 2013). An example of the different stand structures and densities is provided in figures A1 (low density) to A3 (high density) (Canada 2022).

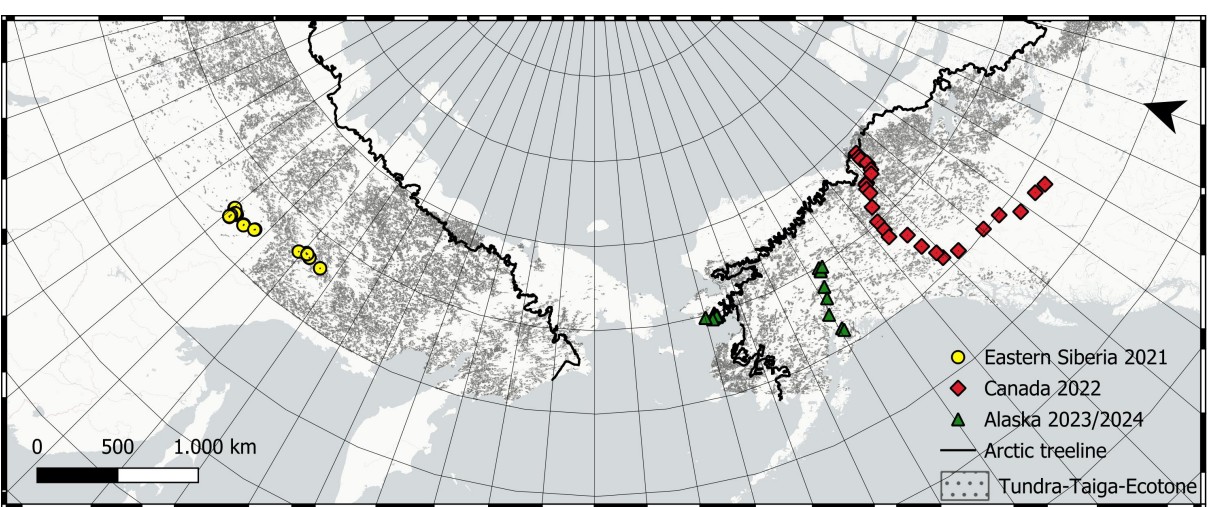

**Figure 2.** Overview of the research area and reference sites from AWI expeditions with available data. Gray hatched areas are covered by forests belonging to the Tundra-Taiga-Ecotone (TTE) derived from the Tree-Canopy-Cover (Ranson et al., 2014). Please note that south of the TTE is the denser boreal Ecotone that is also covered by our expedition sites in Siberia and North America, not visualised in this figure. The black color-coded treeline was defined by the Circum Arctic Vegetation Mapping (CAVM) (Walker et al., 2005); Basemap (Esri©)



**Table 1.** Overview of AWI Expeditions, flight transects, No. of reference plots and data publications

| Expedition | Time frame | No. of point clouds | Flight transects | No. of reference plots | Data publication |
|---|---|---|---|---|---|
| Eastern Siberia 2021 | 01.08.2021 – 06.09.2021 | 72 | EN21201 – EN21262 | 177 | Kruse et al. (2025e) |
| NW Canada 2022 | 10.07.2022 – 19.08.2022 | 25 | EN22002 – EN22071 | 84 | Kruse et al. (2025c) |
| Alaska 2023 | 20.06.2023 – 19.07.2023 | 45 | EN23604 – EN23700 | 109 | Kruse et al. (2025a) |
| Alaska 2024 | 18.06.2024 – 19.07.2024 | 4 | EN24110 – EN24124 | 14 | Kruse et al. (2025b) |

## 2.2 Data acqustion and postprocessing

UAVs (DJI Matrice M300 RTK) were utilised for transect overflights of at least $500 \times 50$ m with acquisitions using a LiDAR sensor (YellowScan Mapper (Table A1), for the 2022 and 2023 flights in addition with RGB camera module). The drone flights were conducted at an altitude of 70 meters with an speed of 5 m/s, resulting in approximately 400 points/m$^2$. The LiDAR mapper followed a predetermined initiation pattern for Inertial Measurement Unit (IMU) initialisation. Precise Point Positioning (PPP) was implemented using coordinates from an EMLID Reach GNSS Basestation and the POSPac UAV software version 8.8 for accurate flight trajectory inertial corrections and georeferencing. For the 2022 to 2024 transects we could also use POSPac created timestamps for the images recorded by the YellowScan RGB module, which are essential for later colorization of the point cloud. Using Yellowscan CloudStation software (version 2206), inaccurate trajectory strips were removed to prevent unrealistic point density from overlapping recordings. Ultimately, the point clouds were generated and linked with the RGB images via their timestamps, enabling effective colorization and enhanced visualization of the collected data. The point clouds from Yakutia 2021 are not colorised. The 3D point clouds of all transects were classified into ground points and above-ground points. Ground classification was performed using the software LASTools (v2.0.2) as described by Isenburg (2023).

Earth System Science Data Discussions — Open Access

## 2.3 Reference plot generation

90

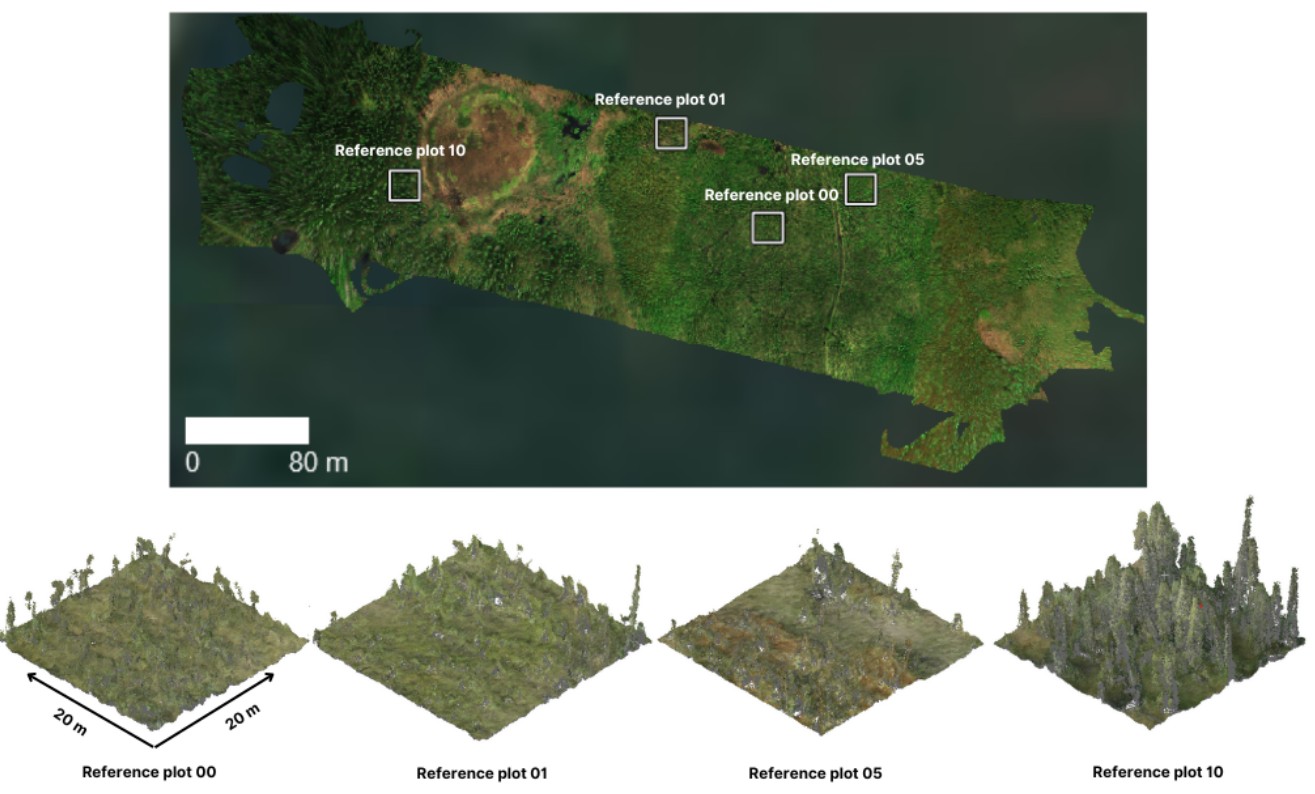

**Figure 3.** Orthomosaic of Site EN23612 - Below are the colorised point clouds of the reference plots representing the site. On plot 00 detailed inventory during field work was carried out. Plots 01 to 10 refering to the size of the trees (01 lowest trees, 10 highest trees)

To create representative forest structure subsets of the point cloud data, we extracted 20 m × 20 m reference plots from the original data sets. A canopy height model (CHM) was generated to identify treetops within the point cloud through a point-to-raster (p2r) algorithm. The heights of the trees served as the basis for determining the locations of the reference plot cut-outs. Trees were categorised by their height from smallest to tallest into 10 subgroups, and one tree from all subgroups, was randomly selected to represent the center of a reference plot. Subsequently, a 20 m x 20 m buffer was established around each selected tree, and the corresponding subsets were cut out. Each reference plot was then exported with a designation corresponding to its respective tree height group (1 to 10), ensuring a systematic representation of tree height variation across the study area. We

95





used all Level 1 (small trees), Level 5 (medium high trees) and Level 10 (tall trees), and in some cases also Level 3 (small to medium trees) and Level 8 (medium to tall trees). Additionally, a fourth 20 m × 20 m cut out was generated if a detailed forest inventory was performed in the field . This plot was then labelled with "00" and acts as the ground reference for the creation of a training data set for species assignment (Fig. 1,3).

## 2.4 Tree segmentation

The segmentation of individual trees, within the 20 x 20 m reference plots, was conducted manually using CloudCompare ((version 2.13) [GPL software] (2025) retrieved from http://www.cloudcompare.org/). Trees that were visually distinguishable by their crowns were extracted and assigned with unique IDs, following a consistent numbering scheme established during ground classification. In this scheme, ID 1 represents unclassified points (including shrubs and unidentified tree parts), ID 2 denotes ground classified points, and individual trees are numbered starting from ID 3. A new scalar field named "Tree" was created to incorporate this numbering system (data type uint64). While this manual approach promised to be more flexible to different forest structures, accurately separating trees can still be difficult. Issues arised when shrub layers were inadvertently included with the trees, or when branches from overhanging trees complicated the segmentation process. The broadleaf forests in the data set were particularly challenging for precise individual tree separation. To ensure good quality segments in the final data set despite these complexities, the process involved multiple quality control mechanisms. We created plots of every tree from two different angles (example Fig. 4) and checked for outlying points.

## 2.5 Manual species assignment

The training data sets are based on the reference plots annotated with "00" where species were documented during the expeditions. Individual trees were manually assigned a species label. Orthorectified images from photogrammetry missions, along with 360-degree panorama images (Kruse et al., 2025d) captured from the ground at plot center, were used for orientation within the point cloud. This information enabled the assignment of species, which were recorded in an additional scalar field labeled "Species" (data type uint8). The numbering scheme for this field continued from the segmentation: ID 1 for unclassified points, ID 2 for ground points and beginning with ID 3, the number coded species. To enhance the species classification model's predictive accuracy, two separate training data sets were created—one for North America and another for Siberia—due to the significant differences in species composition between these regions. For North America, the numbering scheme (Table 2) for species assignment was established to reflect the specific tree species identified in the reference plots, ensuring a tailored approach to data set creation that aligns with regional biodiversity.

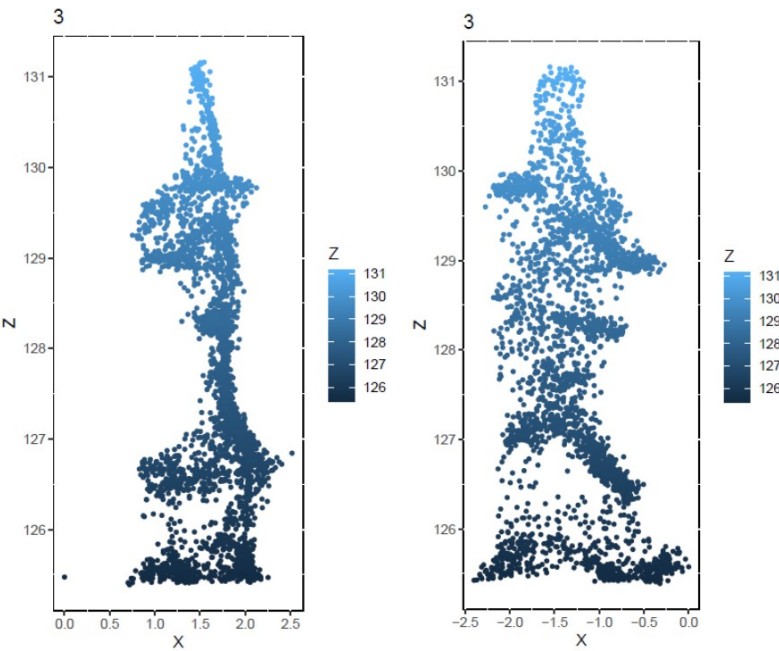

**Figure 4.** Example point cloud of the same tree from two angles, used for quality control. Z is height in m above sea level

**Table 2.** List of tree species with their scalar field IDs, scientific names and abbreviations for the North American data set.

| ID | Scientific Name | Abbreviation |
|----|-----------------|--------------|
| 3  | *Picea mariana* Britton, Sterns & Poggenb. | PIMA |
| 4  | *Picea glauca* (Moench) Voss | PIGL |
| 5  | *Pinus contorta* Douglas ex Loudon | PICO |
| 6  | *Abies lasiocarpa* (Hook.) Nutt. | ABLA |
| 7  | *Larix laricina* (Du Roi) K.Koch | LALA |
| 8  | *Betula papyrifera* Marshall | BEPA |
| 9  | *Populus balsamifera* L. | POBA |
| 10 | *Populus tremuloides* Michx. | POTR |
| 11 | *Picea sitchensis* (Bong.) Carrière | PISI |
| 12 | *Betula neoalaskana* (Regel) Ashburner & McAll. | BENE |
| 13 | *Alnus glutinosa* L. | ALGL |

In North America trees of the *Picea* genus are the most abundant across most of the sites, the same is true for *Larix spp.* in Yakutia. To balance this, the training data set were equalised to a maximum of 50 individuals per species. In Yakutia, the species composition of the boreal zone differs from the one found in North-America. The most dominant genus is *Larix* Mill.,



**Table 3.** List of tree species with their scalar field IDs, scientific names and abbreviations for the Yakutian data set.

| ID | Scientific Name | Abbreviation |
|----|-----------------|--------------|
| 3 | *Larix* spp. | LARIX |
| 4 | *Pinus sylvestris* (Thunb.) | PISY |
| 5 | *Betula* spp. | BETU |

represented by the species *L. sibirica* Ledeb. and *L. gmelinii* Rupr. While both species are found in different parts of Siberia, their appearances hardly differ (Schulte et al., 2022). Thus, we focus on the genus level for *Larix spp.* since a separation of the species based only on structural parameters is not to be expected. The same is true for *Betula pendula* Roth and *Betula platyphylla* Sukaczev, therefore we decided for *Betula spp.*. The numbering scheme for Yakutia is presented in Table 3.

### 2.6  Extracting tree structural and spectral variables

A total of eleven structural and two spectral variables (Table 4) were calculated to train the first randomForest classifier for the North American data set. The spectral variables were derived from the colorization of the point cloud, which only includes the values the visible spectrum of light. Given that only the three visible light bands (Red, Green, Blue) are available, we calcuated the pseudo NDVI known as Normalised Green-Red Difference Index (NGRDI) (Tucker, 1979) and the Visible Atmospherically Resistant Index (VARI) (Gitelson et al., 2002) (Table 4).

**Table 4.** Summary of variables and their equations

| Variable | Equation / Source | Abbreviation |
|----------|-------------------|--------------|
| Normalised green red difference index | $\frac{\text{GREEN}-\text{RED}}{\text{GREEN}+\text{RED}}$ | NGRDI |
| Visible atmospherically resistant index | $\frac{\text{GREEN}-\text{RED}}{\text{GREEN}+\text{RED}-\text{BLUE}}$ | VARI |
| Pointedness coefficient | $\frac{\max(Z)-\text{mean}(Z)}{\max(Z)}$ | pointedness |
| Crown relief ratio | $\frac{\text{mean}(Z)-\min(Z)}{\max(Z)-\min(Z)}$ | CRR |
| Coefficient of variation of height | $\frac{\text{sd}(Z)}{\text{mean}(Z)}$ | CV_Z |
| Top angle | derived from fitted triangle (Fig. 5) | top_angle |
| Widest distance within the tree crown on a horizontal plane | derived from fitted triangle (Fig. 5) | widest_distance |
| Relative height of widest part of the crown | derived from fitted triangle (Fig. 5) | relative_height_widest |
| Volume of concave hull | derived from alphashape3d package in R | vol_concave |
| Density of points within concave hull | total number of points divided by vol_concave | density_concave |
| Height of the 99.9% quantile | 0.999 quantile of $Z$ | ZQ999 |
| Height of the 99% quantile | 0.990 quantile of $Z$ | ZQ99 |
| Vertical variability of points on z axis | standard deviation of $Z$ | vertical_variability |

In addition to spectral indices, we calculated 11 structural parameters to quantify tree shapes (Table 4). These include the pointedness coefficient, the crown relief ratio (CRR) and the coefficient of variation of height (CV_Z), calculated as shown in Table 4. The vertical variability, defined as the standard deviation of tree height (Z), and the 99.9th percentile (Zq999) were also included. We assessed point density and volume within the concave hull, utilizing the alphashape3d (Fig. 5a) package in R, following the methodology proposed by Irwin et al. (2024) and Vauhkonen et al. (2009). Inspired by Qian et al. (2023),

who fitted geometric shapes into tree crown point clouds to differentiate species, we fitted triangles from the widest part of the crown to the top of the tree using a 2D projection of the trees point cloud (Fig. 5b). This approach allowed us to derive further structural parameters: the top angle of the triangle, the length of the lower leg, and the height of the lower leg relative to tree height. These parameters (Fig. 5b) are intended to quantify whether the tree crown is wide or narrow and to identify the height at which the widest part of the crown is located. These metrics were extracted for every tree that was assigned a

species manually. The resulting data set, with two spectral and 11 structural variables, is the basis for training the randomForest classifier for North America. The point clouds from Yakutia 2021 do not contain the RGB data, and neither do six point clouds from Canada 2022. These were instead predicted using a randomForest classifier that was based only on structural parameters.

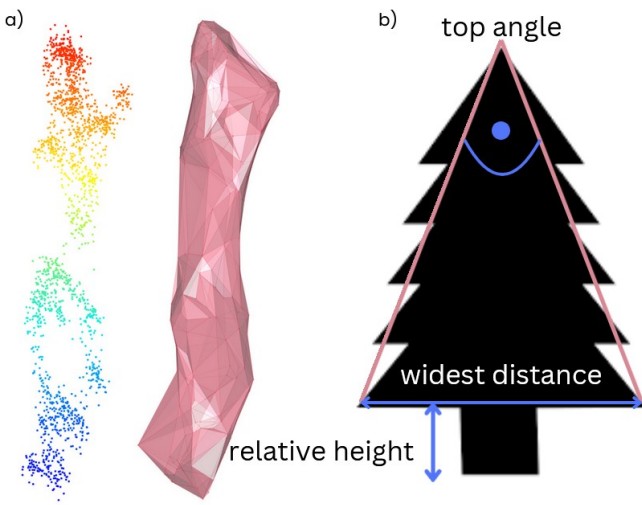

**Figure 5.** a) Example point cloud of individual tree and alphashape (alpha = 1) b) Graphical representation of the fitted triangle for extracting the widest distance its relative height and top angle

## 2.7    Modeling and Prediction

Due to variations in data availability (with and without RGB) and differences in species composition across biomes such as Yakutia, the Seward Peninsula, and continental North America, we had to develop four randomForest classifiers (Fig. 6) based on distinct training data subsets. Two models utilised structural variables exclusively from point clouds in Yakutia (RF4) and six non-colorised point clouds from Canada (RF3) due to missing RGB data. The latter was trained using the structural information



from all point clouds recorded in Canada. The other two models incorporated RGB information along with related variables
(VARI, NGRDI) to predict data from continental Canada and Alaska (RF1), and coastal regions of the Seward Peninsula
(RF2). The classifiers were constructed using the caret package in R (v 6.94 (Kuhn 2008)), employing 10-fold cross-validation
to assess model performance. Hyperparameter tuning was performed to determine optimal values for mtry and ntree (mtry
equals the number of variables tried at each split, ntree equals the number of decision trees built by the model).

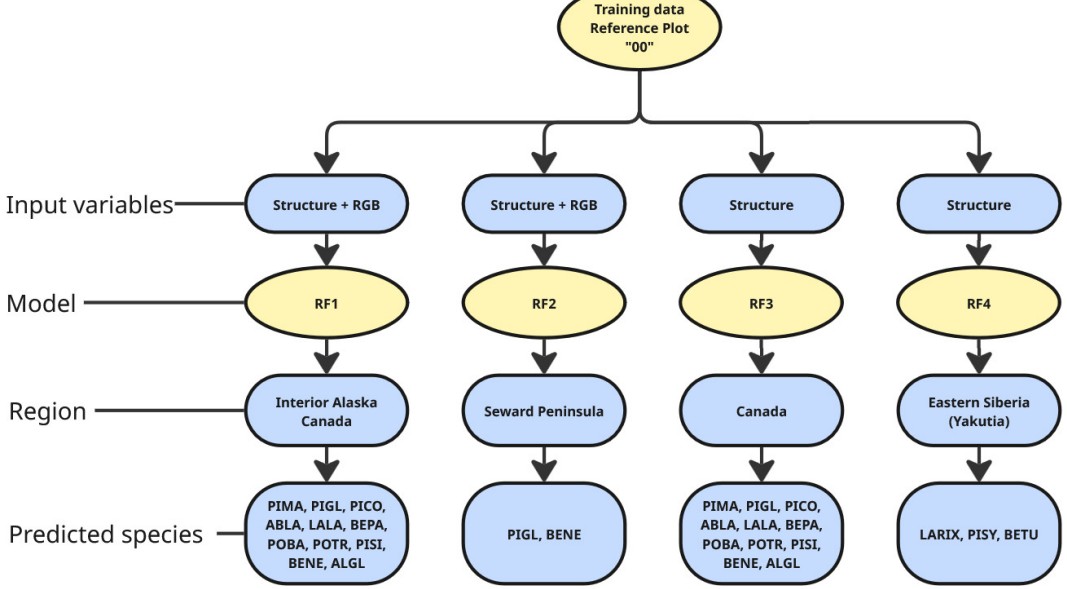

**Figure 6.** Schematic representation of the four used randomForest classifiers (RF1-RF4). Depicted are the variables used to train and the
region with corresponding species they were used for. (Tree species abbreviations are described in table 2 for North American tree species
and table 3 for Siberian tree species)

The four different randomForest models were employed to predict the species of all segmented trees within the point clouds
of the corresponding region. We added two scalar fields: "Species", which contains a numeric value representing the predicted
species, and "Probability," which ranges from 0 to 1 to indicate prediction certainty (1 = perfect match). Predictions for various
regions in North America were manually modified based on regional biodiversity. For instance, RF1 was used for Alaska and
Canada, where predictions such as *Betula neoalaskana* in Canada were reassigned to *Betula papyrifera* due to local biodiversity
considerations. This supervised approach aimed to enhance data set quality, as the model sometimes struggles to distinguish
between species of the same genus. The following prediction translations were applied:

– *Betula neoalaskana* in Canada was translated to *Betula papyrifera*

– *Picea sitchensis* in Canada was translated to *Picea* genus with highest probability

– *Abies lasiocarpa* in Alaska was translated to *Picea* genus with highest probability





A file with metadata was created for every point cloud file, including the structural and (if applicable) spectral variables and prediction probability for every single tree on which the prediction was made.

# 3 Results

## 3.1 Representativness of BorFIT Plots

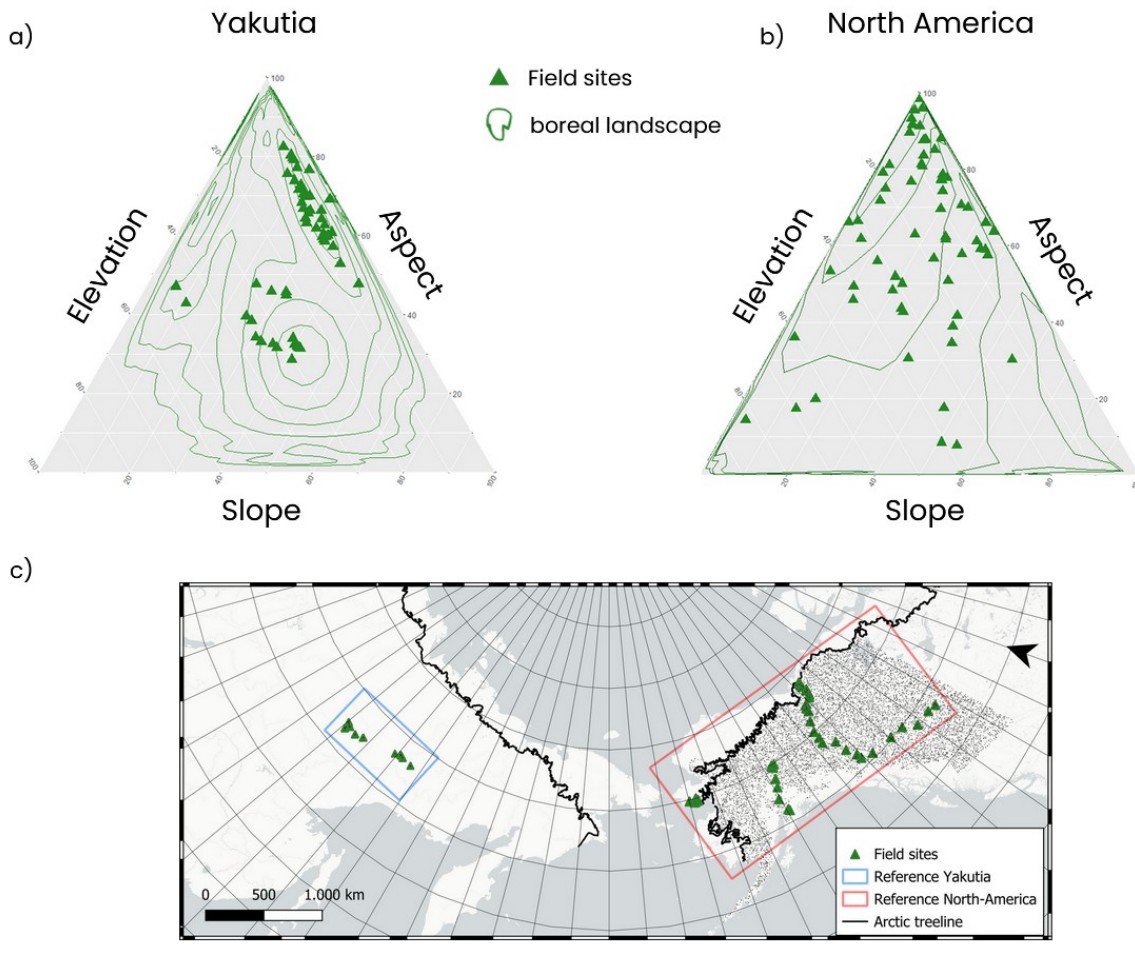

**Figure 7.** Ternary plots of field sites within the diversity range of a) Yakutian and b) North American boreal forest topography (contour density lines) for elevation between 0 to 3141 m asl (North America) and 65 to 2572 m asl (Yakutia), slope angles of 0 to 53° (North America) and 3 to 79° (Yakutia) and all ranges of aspect (Ensemble Digital Terrain Model, EDTM(Ho, Hengl, and Parente, 2023)) within the boundaries depicted in the c) reference map

BorFIT is meant to be a training data set for AI applications. However, the data offer valuable information on forest structure across our North-American and Yakutian research area. Our sites cover both lowland and mountainous regions and are



thereby representative for a broader boreal region. Especially in North America, our sites are evenly spread across the present topographic conditions, while the Yakutian sites are more clustered in lowland areas with only a few sites located in the mountains, and not covering the Northern taiga-tundra biome (Fig. 7). To avoid including poor classification results, we sub-
setted the data for the following data visualisation (Fig. 8 onward) using only trees with a prediction probability of over 0.8, which significantly reduces the number of trees included to 1497 for the Yakutia data set and 3060 for the North American point clouds. The most abundant species in our North American data set is *Picea glauca* followed by *Picea mariana* while the data set in Yakutia is dominated by *Larix spp.* (Fig. 8). This is in agreement with previous studies on boreal forest biodiversity (Ivanov et al., 2023; Xi et al., 2024). The predicted species distributions with environmental variables in Bor-
FIT reveals patterns consistent with our field observations: in Yakutia, warmer lowlands are inhabited by *Pinus sylvestris* in the sandy soils of the Eastern Lena river bank, *Larix* spp., and *Betula* spp., while colder mountainous areas are dominated by *Larix* spp.; in North America, northern regions are dominated by *Picea* spp., often in mixed stands, and broadleaf trees like *Populus balsamifera* and conifers such as *Abies lasiocarpa* are more common in the mixed forests of, for example, British Columbia in Canada —patterns that are reflected in the BorFIT predictions against environmental factors (Fig. 9).

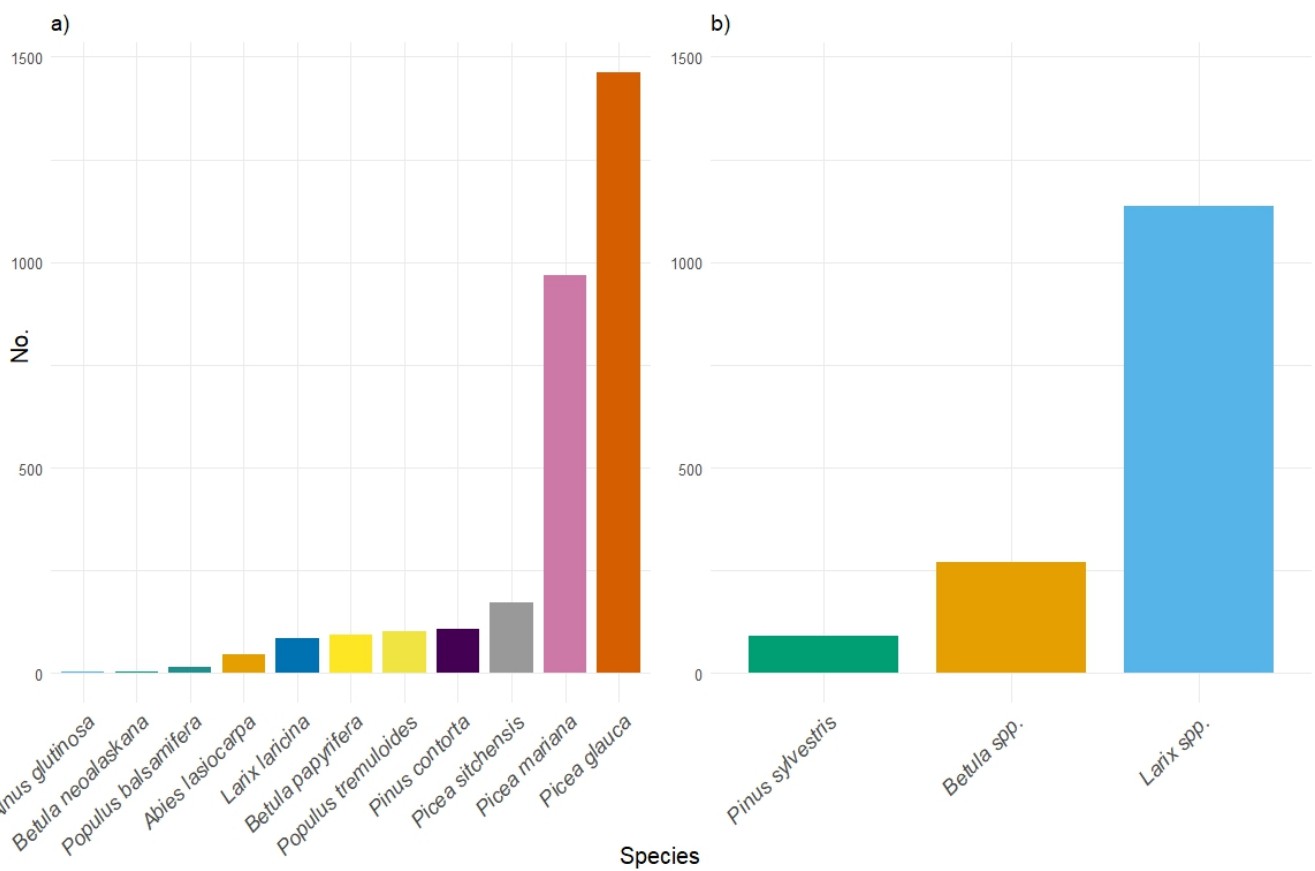


**Figure 8.** Predicted abundance (number of trees) of different tree species in the BorFIT data set. a) North America (n = 3060); b) Yakutia (n = 1497) (subsets with prediction probability > 0.8)



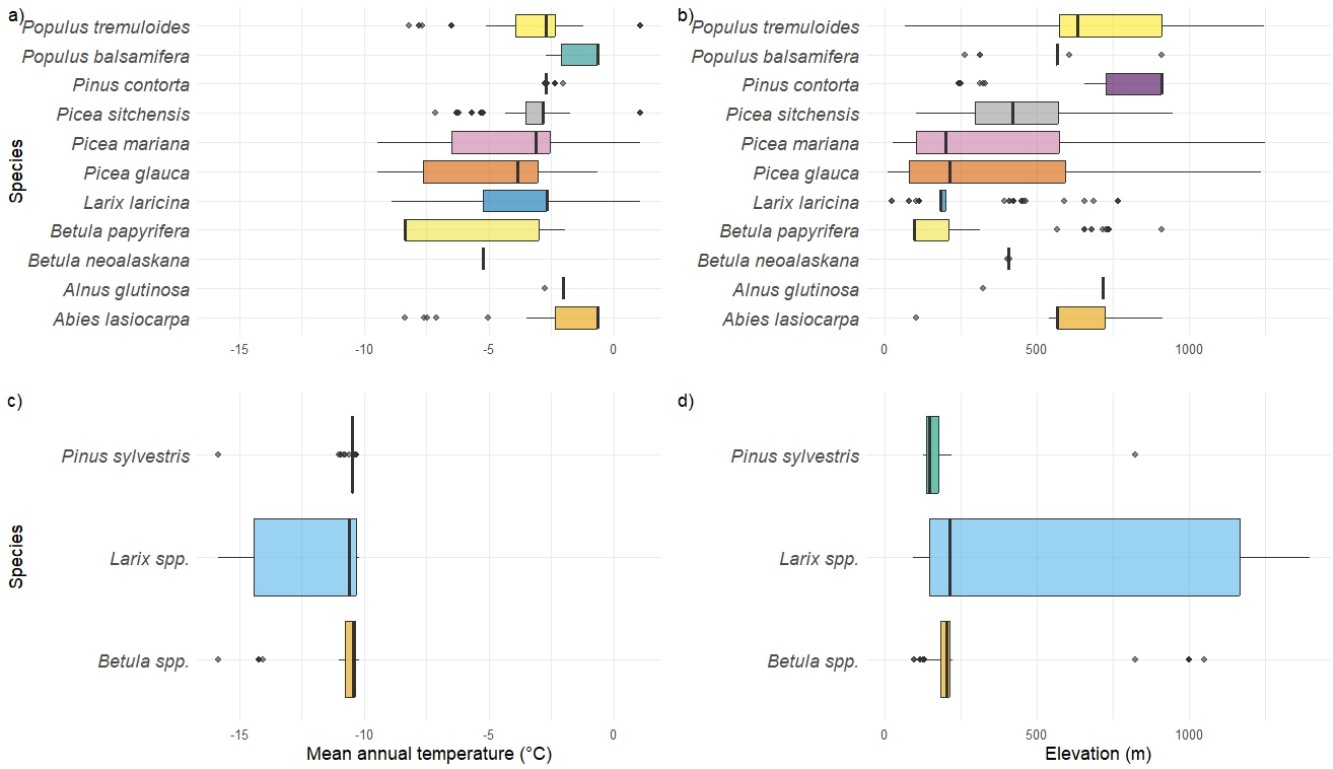

**Figure 9.** Boxplots of species distribution against environmental conditions. a) North American species against mean annual temperature (MAT); b) North American species against elevation; c) Yakutian species against MAT; d) Yakutian species against Elevation; MAT derived from WorldClim (Fick and Hijmans, 2017) ; Elevation according to European Space Agency (2024); (subset with prediction probability > 0.8)

## 3.2 Classification performance

Four tailored classifiers were trained to account for the diverse localities included in the BorFIT data set. Table 5 gives an overview of the different models performance metrics. The mean accuracy across the 10 folds that were used, the out-of-
bag (oob) error, as well as the values for mtry and ntree, defined through hyperparameter tuning, are indicated. While most species predictions had class errors between 0% and 26%, certain species, such as *Betula neoalaskana* and *Alnus glutinosa*, presented greater challenges, with errors reaching up to 58%. This increased error may be due to their limited representation in the training data. A full overview of the models class errors is provided in the Appendix (Table A2 to A5). Especially RF3 struggled with several species since it had the highest number of classes while only relying on structural information
(Table A4). The models feature importances (Fig. 8) indicate how the models distinguish between species. Some species share similarities across multiple attributes, making differentiation more difficult and increasing uncertainty. Notably, various *Betula* and *Picea* species share structural similarities and are primarily differentiated by their spectral attributes (Fig. 12). Overall, 16,530 trees were predicted with a mean probability of 0.67.



**Table 5.** Overview of the randomForest models' performances and data (mtry is the number of variables tried at each split, ntree is the number of decision trees built by the model, OOB refers to out of bag, the internal validation metric for model performance (the lower the better) (Overview on spectral and structural variables in table 4. The abbreviations of tree species are shown in table 2 for North America and table 3 for Siberia)

| Model | Region | Variables | Species Codes | mtry | ntree | Accuracy in % | OOB Error in % |
|---|---|---|---|---|---|---|---|
| RF1 | Interior Alaska, Northwestern Canada | 11 structural variables + VARI + NGRDI | PIMA, PIGL, PICO, ABLA, LALA, BEPA, POBA, POTR, PISI, BENE, ALGL | 12 | 500 | 86 | 20 |
| RF2 | Seward Peninsula | 11 structural variables + VARI + NGRDI | PIGL, BENE | 7 | 500 | 91 | 21 |
| RF3 | Northwestern Canada | 11 structural variables | PIMA, PIGL, PICO, ABLA, LALA, BEPA, POBA, POTR, PISI, BENE, ALGL | 2 | 500 | 73 | 42 |
| RF4 | Eastern Siberia (Yakutia) | 11 structural variables | LARIX, PISY, BETU | 2 | 500 | 79 | 20 |

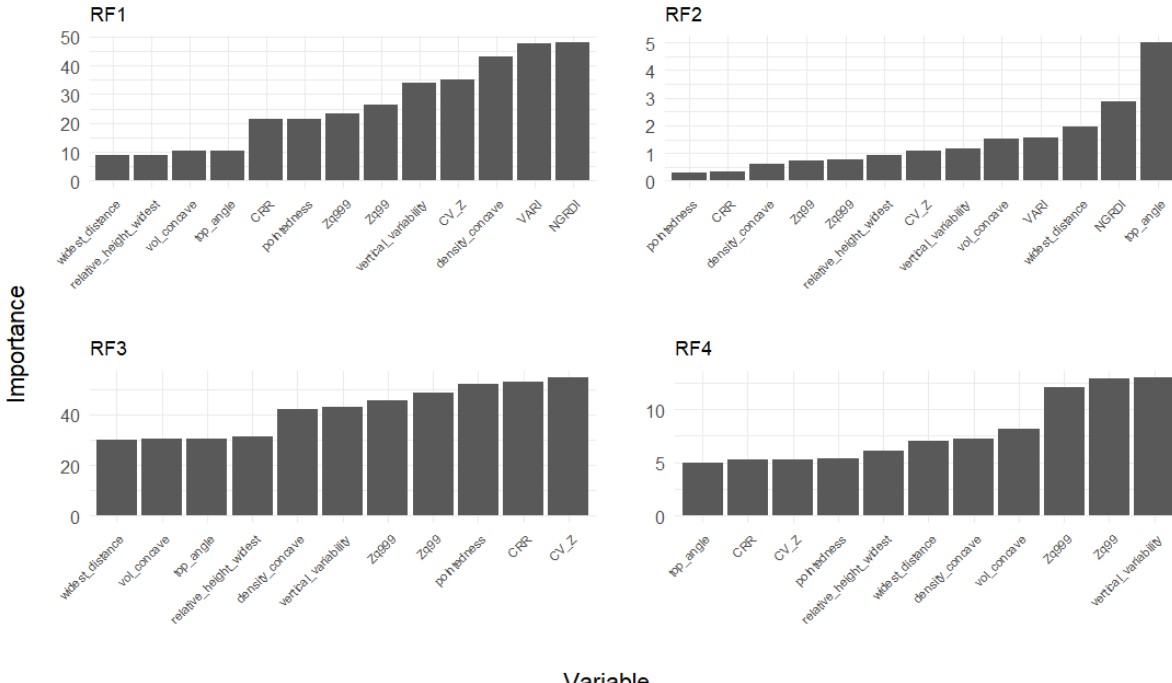

**Figure 10.** Variable importance of the 4 randomForest classifiers. Notably the models with RGB (RF1, RF2) relied heavily on spectral variables; top_angle was most important for RF2 which only decided between *Picea sitchensis* and *Betula neoalaskana*, and the shape of the crown was the deciding factor.



### 3.3 Data set Properties

The primary objective of the BorFIT project was to create a comprehensive database of individual tree point clouds from the boreal biome. We successfully segmented 16,530 trees across 14 taxa. While the number of reference plots is similar in Yakutia (N=177) and North America (N=208), the segmented tree counts differ significantly: over 10,000 trees were segmented in North America compared to only 6,085 in Yakutia. This discrepancy is attributed to the less dense stands of taller *Larix spp.* trees in Yakutia, compared to the denser populations of *Picea mariana* found near the arctic tree line in Canada and Alaska,

where up to 200, mostly shorter, individuals can grow within a 20 m x 20 m reference plot. The smallest trees in our data set are approximately 50 cm tall.

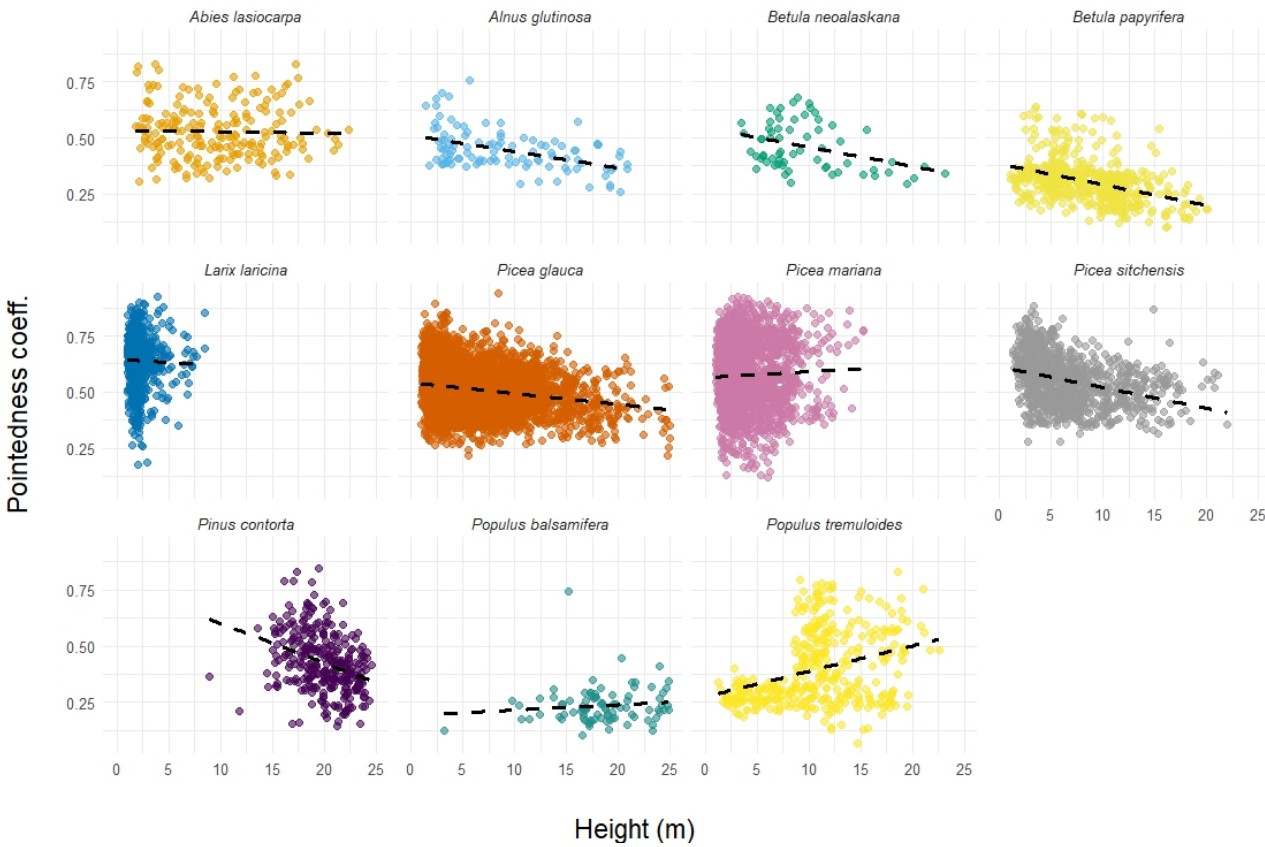

**Figure 11.** Height vs. pointedness coefficient (Table 4) of North American species (subset with prediction probability > 0.8)

This is due to difficulties in the segmentation process, when distinguishing between young trees and shrubs. According to our data set, in North America the mean tree height is 6.3 m, in Yakutia it is 8.3 m; while some individuals in North America reach

over 30 m, while the maximum in Yakutia is below 25 m. The randomForest classifier was trained to identify structural variables



of tree species regardless of their growth stage; however, mature individuals of, e.g. *Picea mariana*, exhibit fundamentally different structures compared to younger trees, complicating the learning process across most species.

**Figure 12.** Density of species' distribution in North America against structural and spectral characteristics a) predicted species against NGRDI; b) predicted species against VARI; c) predicted species against top angle; d) predicted species against coefficient of variation of height; e) predicted species against CRR; f) predicted species against pointedness coefficient

**Figure 13.** Density of species' distribution in Eastern Siberia against structural and spectral characteristics a) predicted species against Vertical variability; b) predicted species against alphashape volume; c) predicted species against pointedness coefficient; d) predicted species against crown relief ratio

The relationship between the height and width of BorFIT trees changes as they grow. Some species (*Alnus glutinosa, Betula neoalaskana, Betula papyrifera, Picea glauca, Picea sitchensis, Pinus contorta*) become less pointed with size, while others (*Picea mariana, Populus balsamifera, Populus tremuloides*) become more pointed (Fig. 11). This variability complicates species classification by the randomForest classifier. In addition, similarities in structural and spectral traits between species make accurate classification even more diffi-



cult. In contrast, figures 12 and 13 highlight the distinct differences that allowed robust classification results.

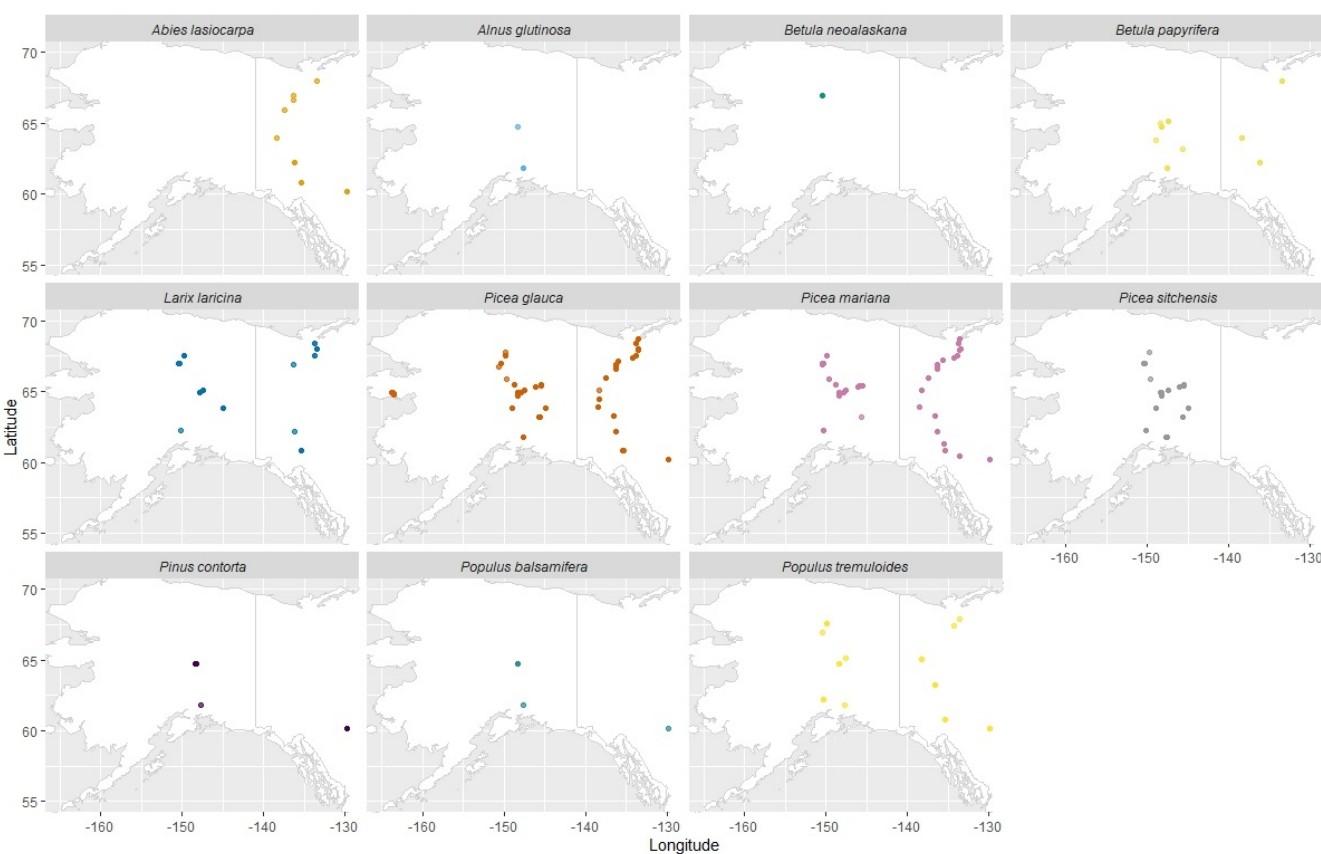

**Figure 14.** Predicted tree species distribution in North America; data subset (probability>0.8)

The predicted distribution of North American BorFIT trees is mapped in Fig. 13. Clearly visible is the dominance of the *Picea* genus in Alaska and Northwestern Canada within our data set. Other species, e.g *Betula neoalaskana* and *Alnus glutinosa* are underrepresented. This is due to the subset of the data set that was used for the mapping (Probaility > 0.8) since these species

yielded higher class errors during prediciton (Table A2 - A5).



## 4 Discussion

### 4.1 Manual segmentation

The segmentation of over 16,000 individual trees across diverse boreal regions provides a unique data set to assess forest structure and species composition at novel scales, in Alaska, Northwestern Canada and Eastern Siberia. Segmenting individual trees from forest point clouds remains a bottleneck for retrieving stand structure and species distribution patterns. Studies that produced data sets that contain similar quantities of individual tree point clouds have focused either on local maximum filtering for tree segmentation (Dubrovin et al., 2024) (3,600 trees) or on various segmentation techniques, including canopy height based techniques (Puliti et al., 2025) (20,000 trees). Segmentation based on 2D representations of the canopy tends to overlook trees when dealing with overlapping crowns (Brieger et al., 2019). More advanced techniques based on deep learning require suitable trainig data sets, tailored towards the respective tree and stand structure (Chehreh et al., 2023). Regardless of the method used, high quality segmentation often requires manual refinement (Puliti et al., 2025). Therefore, the segmentation of BorFIT trees was performed manually to acquire high quality segments without requiring a tailored training data set. This process is labor-intensive and subject to errors, particularly in dense broadleaf forests where overlapping canopies and understory vegetation confound structural and spectral feature extraction. Measures for quality control (as mentioned in 2.4) are compulsory to ensure an accurate segmentation result.

### 4.2 Tree species classification performance

The robust performance of our models (mean accuracy 82%) can be attributed to the efforts invested in manual segmentation of trees and tailoring different models for distinct biomes. These scores are consistent with previous studies employing similar methodologies (Persson A., 2004; Shi et al., 2018; You et al., 2020). Michałowska and Rapiński (2021) reviewed studies discriminating tree species using attributes derived from LiDAR point clouds with various algorithms including randomForest, noting that overall accuracy tends to decrease as the number of species to distinguish increases. The maximum number of species in any reviewed study was six, yielding overall accuracies between 0.57 and 0.79. In contrast, RF1 from our study was trained on 11 different species and achieved an overall accuracy of 0.86, supported by the inclusion of spectral variables which ranked among the most important during classification. RF3, which excluded spectral attributes but still differentiated 11 species, resulted in an overall accuracy of 0.73, a performance that remains robust compared to the studies reviewed by Michałowska and Rapiński (2021). RF2, classifying only two species with spectral variables, achieved the highest accuracy of 0.91, as expected, while RF4 classified three species using only structural variables with an accuracy of 0.79, falling within the 0.62 to 0.95 accuracy range reported by Michałowska et al. (2021) for similar scenarios. Underrepresentation of species such as *Alnus glutinosa* and *Betula neoalaskana* in the training data led to reduced classification accuracy for these taxa, highlighting a common challenge in ecological remote sensing: scarcity of annotated data for less common species. This limitation, well-documented in machine learning literature (Jafarigol and Trafalis, 2023), shows that insufficient or imbalanced datasets can significantly hinder model generalizability and performance, as reflected by considerably lower class accuracies for species with fewer training samples. Furthermore, our results emphasize the importance of spectral features in distinguishing tree





species; RF3, relying solely on structural data, produced lower overall accuracy and higher out-of-bag errors compared to models including spectral variables. These findings underscore the critical role of balanced training datasets and the integration of spectral and structural information to achieve robust species classification. In conclusion, our classification yielded robust results, providing users with a high-quality product for the analysis of spectral and structural traits of boreal trees.

### 4.3 Predicted species distribution

Some broadleaf species are underrepresented in the data subset (probability > 0.8), leading to irregularities in their predicted distribution. *Populus tremuloides* and *Picea sitchensis* have been predicted in point clouds sourced from regions far north of the Arctic Circle where the species have not been observed during field work. *P. mariana* and *P. glauca* are the most dominant species in our North American study region, often found in mixed stands, this pattern is reflected in their predicted distribution. However, Roland et al. (2013) report specific site preferences for the two species (*Picea mariana* dominating at lower elevation compared to *Picea glauca*). These are not clearly visible within our data. The low class errors in RF1 when predicting these species compared to RF3 suggest that spectral attributes were important for the differentiation. The lower performance of RF3 could therefore be masking these patterns. However, most of the distribution patterns visible in Figs. 9 (against environmental variables) and 13 (spatial) align with our experiences during field work. A clear dominance of *Picea spp.* in North America is visible. The Same is true for *Larix spp.* in Eastern Siberia while the other species are bound to warmer regions (*Populus tremuloides, Populus balsamifera, Abies lasiocarpa, Pinus sylvestris, Betula spp.*) compared to their respective dominant competitors. The BorFIT training data set derived from LiDAR (and RGB) delivers robustly classified tree species across the Northern boreal region that can be used for further AI classification of tree species from LiDAR point clouds. However, it should be noted that the overall plausible pattern visible in our data is highly linked to sufficient samples in the training data and the availability of spectral attributes.

### 4.4 BorFIT training data set

BorFIT bridges the gap between small-scale individual tree assessments (Wittke et al., 2024) and large-scale remote sensing products for stand structure analysis (van Geffen et al., 2022). While the FOR-instance data set (Puliti et al., 2023) provides a global benchmarking and training resource for AI-based tree segmentation—comprising segmented individual trees in point clouds from various regions, the boreal trees included are from Europe only, and all trees lack species annotations. In contrast, BorFIT is the first data set of its kind to cover North American and Siberian boreal regions with species-level classification, filling a critical gap for large-scale, species-resolved analyses in these areas (Puliti et al., 2023; Dubrovin et al., 2024; Puliti et al., 2025). Species-annotated tree point clouds like BorFIT enable a wide range of advanced forestry and ecological applications. They can be used to support the development of AI-driven monitoring tools for forest health, biomass estimation, and carbon stock assessments. Such data sets also facilitate studies of species-specific growth patterns, competition, and responses to environmental change, enhancing our understanding of forest dynamics and biodiversity.



## 5 Data availability

The BorFIT data set is published on the PANGAEA https://doi.org/10.1038/s41597-023-02269-x (Felden et al., 2023) data repository and is available for download https://doi.pangaea.de/10.1594/PANGAEA.980505 (Kruse et al., 2025f).

BorFIT is based on four point cloud data collections from AWI expeditions which are also published on PANGAEA as follows:

– Point clouds from AWI expedition 2021 to Yakutia (Kruse et al., 2025e) https://doi.pangaea.de/10.1594/PANGAEA.980735

– Point clouds from AWI expedition 2022 to Northwestern Canada (Kruse et al., 2025c) https://doi.org/10.1594/PANGAEA.977771

– Point clouds from AWI expedition 2023 to Alaska (Kruse et al., 2025a) https://doi.pangaea.de/10.1594/PANGAEA.980485

– Point clouds from AWI expedition 2024 to Alaska (Kruse et al., 2025b) https://doi.pangaea.de/10.1594/PANGAEA.980757

The R code used for training the randomForest classifier and predicting the species will be made publicly available on Zenodo and GitHub (https://github.com/StefanKruse/BorFIT/tree/main)

## 6 Conclusion

We introduce a novel, large-scale boreal forest tree species reference data set enabling extraction of structural and spectral (RGB) parameters for 14 tree taxa. The data set's diversity in bioclimatic and topographic conditions, combined with robust species prediction results, supports representative analyses of boreal forest composition and structure. BorFIT enables the digital representation of 3D forest structure which is representative of forest condition, functionality, biodiversity and evolution (Hall et al., 2011). 3D forest structure can be upscaled using spaceborne LiDAR (e.g., GEDI, ICESAT-2) or synthetic aperture radar (SAR). Thus, improving the accuracy of large-scale forest structure and composition landcover maps. Limitations include restricted spectral information from RGB sensors and lower prediction accuracies for species with either structural/spectral similarities to other taxa or low representation in the training data, such as *Alnus glutinosa*, *Picea sitchensis*, and *Betula neoalaskana*. Despite these challenges, the data set produced plausible species predictions and offers substantial value for future research. Integrating additional data sources, such as hyperspectral or full waveform LiDAR, could further enhance classification accuracy, as suggested by other studies (Shi et al., 2018; Axelsson et al., 2018; Heinzel, 2012; Korpela et al., 2023).

In summary, advanced technologies like LiDAR and AI, underpinned by comprehensive data sets, are essential for effective boreal forest monitoring. These tools not only facilitate carbon stock assessments but also provide critical information for evaluating the impacts of climate change on species distribution patterns.



330  *Code availability.* The R code used for training the randomForest model and predicting the species will be made publicly available on Zenodo and GitHub (https://github.com/StefanKruse/BorFIT/tree/main). The data is uploaded to PANGEA (Kruse et al., 2025f). Included are the point cloud files, one for each of the 384 reference plots. Additional material will be supplied, including the point clouds with manually assigned species that were used for creating the training data for randomForest and a master table which acts as an overview of the included files.

335  **Appendix A**

**A1**

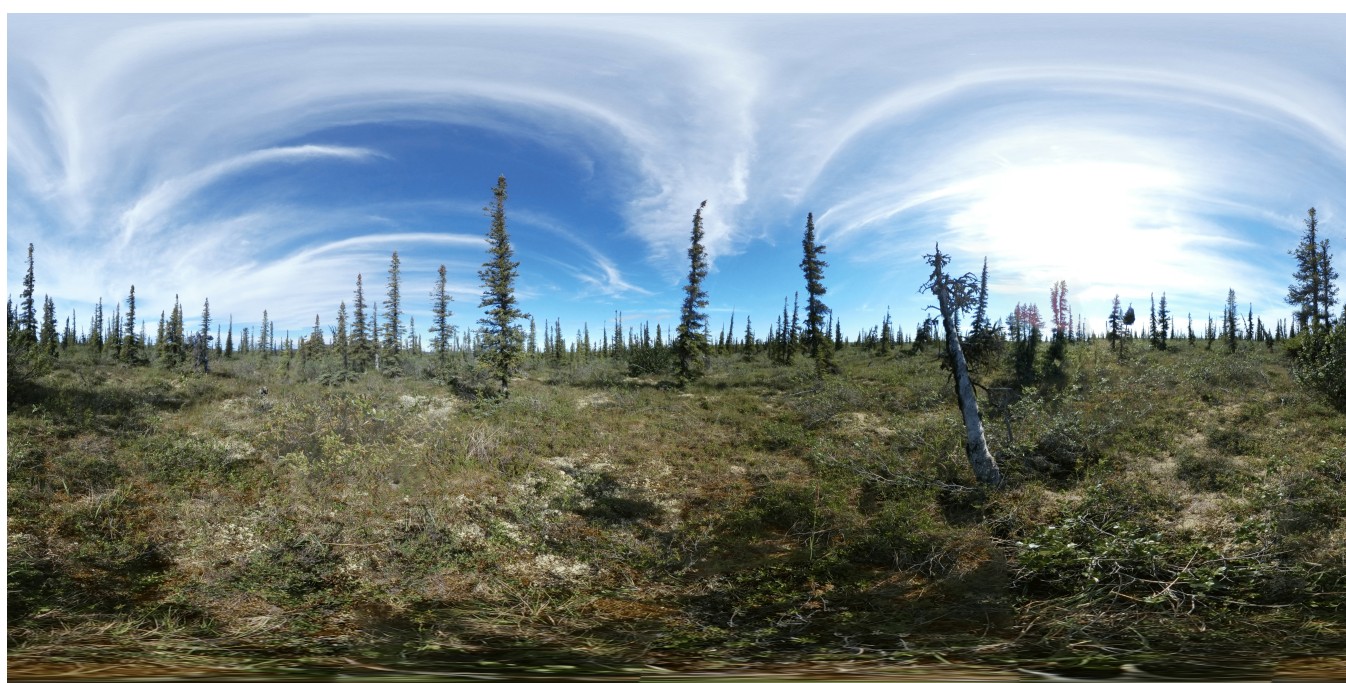

**Figure A1.** Panoramic image of field site EN22006 north of Inuvik (Northwestern Territories, Canada)

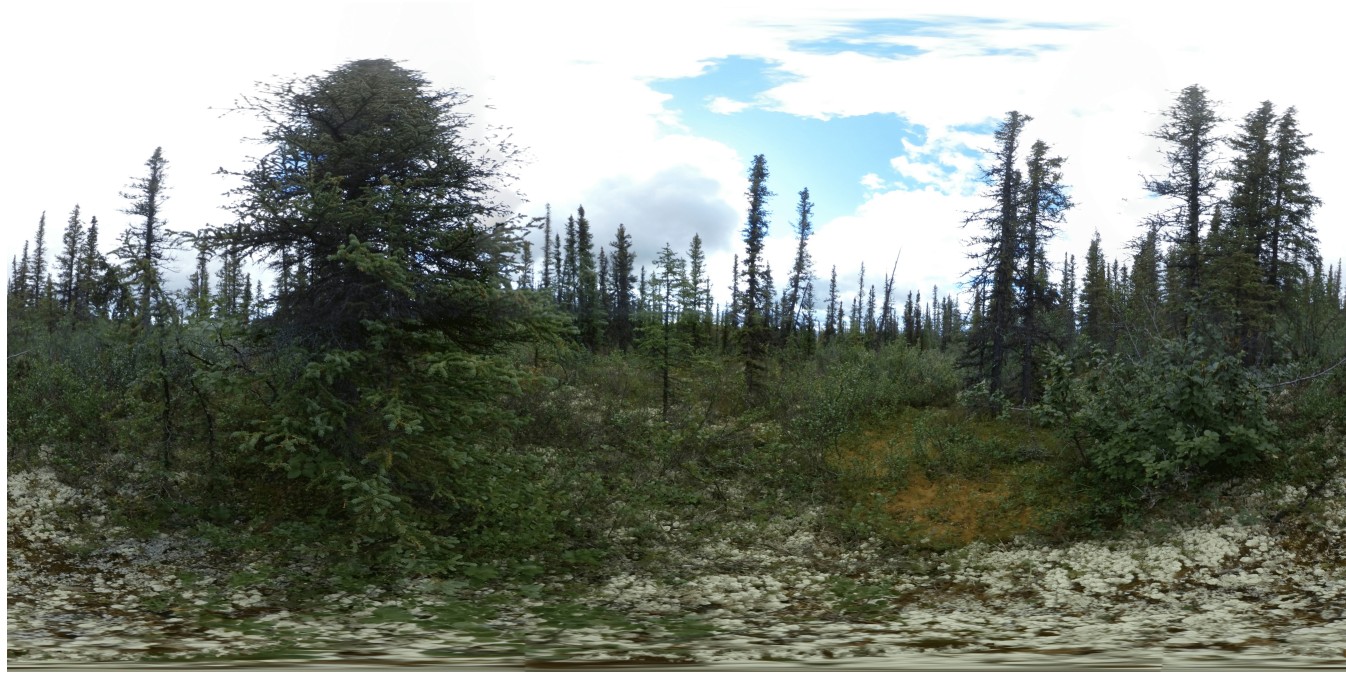

**Figure A2.** Panoramic image of field site EN22033 south of Fort McPherson (Northwestern Territories, Canada)

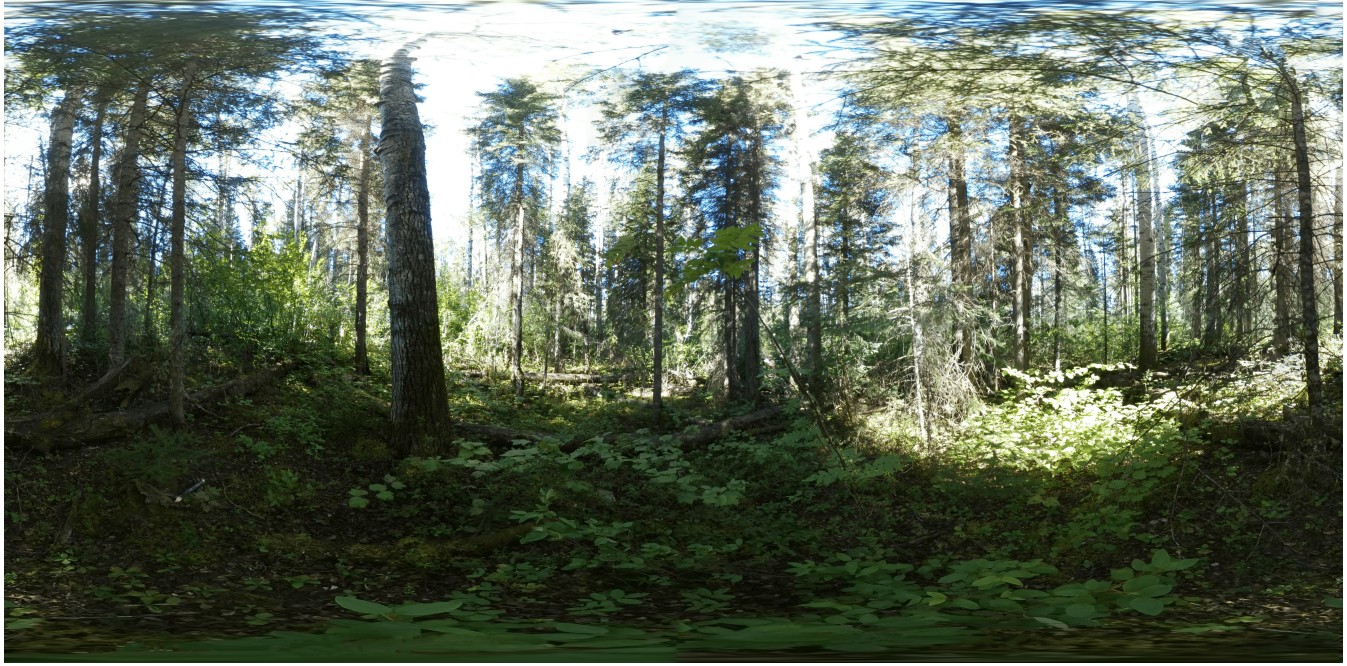

**Figure A3.** Panoramic image of field site EN22071 near Fort Nelson (British Columbia, Canada)



340  **A1**

**Table A1.** Technical properties of the YellowScan Mapper

| Property | Value |
|---|---|
| Scanner type | Livox Horizon |
| GNSS IMU type | Applanix APX-15 UAV |
| Weight [kg] | 1.4 |
| Pulse Rate [Hz] | 240,000 |
| Field of View [°] (horizontal / vertical) | 81.7 / 25.1 |
| Laser Wavelength [nm] | 905 |
| Beam Divergence (horizontal / vertical) [mrad] | 4.8 / 0.5 |
| Accuracy / Precision | 20 / 30 |
| Returns max. | 2 |





**Table A2.** Class Errors for RF1

| Class | Class Error |
|-------|-------------|
| ABLA | 0.115 |
| ALGL | 0.000 |
| BENE | 0.125 |
| BEPA | 0.116 |
| LALA | 0.333 |
| PICO | 0.082 |
| PIGL | 0.264 |
| PIMA | 0.000 |
| PISI | 0.175 |
| POBA | 0.122 |
| POTR | 0.167 |

**Table A3.** Class Errors for RF2

| Class | Class Error |
|-------|-------------|
| BENE | 0.000 |
| PIGL | 0.091 |

**Table A4.** Class Errors for RF3

| Class | Class Error |
|-------|-------------|
| ABLA | 0.207 |
| ALNUS | 0.333 |
| BENE | 0.417 |
| BEPA | 0.364 |
| LALA | 0.389 |
| PICO | 0.077 |
| PIGL | 0.537 |
| PIMA | 0.220 |
| PISI | 0.708 |
| POBA | 0.075 |
| POTR | 0.545 |

**Table A5.** Class Errors for RF4

| Class | Class Error |
|-------|-------------|
| BETU | 0.216 |
| LARIX | 0.182 |
| PISY | 0.214 |

*Author contributions.* According to CRediT:

S. Kruse: Conceptualization-Lead, Investigation, Resources, Writing – Review  Editing, Supervision, Project administration, Funding acquisition

J. Schladebach: Conceptualization, Investigation, Methodology, Formal Analysis, Data Curation, Writing – Original Draft,Writing – Review and Editing, Supervision

L. Enguehard: Data Curation

J. Broers: Data Curation

J. Tretton: Data Curation

345



A. Gorshunova: Data Curation

350  M. Wieczorek: Data Curation

R. Jackisch: Data Curation

J. Gloy: Data Curatuion

K. Hao: Data Curation

B. Heim: Conceptualization, Resources, Writing – Review  Editing

*Competing interests.*  Author B. Heim is a member of the editorial board of the journal.

*Acknowledgements.*  The project was funded by the DataHub Information Infrastructure funds, project BorFIT. Additional funding for equipment was provided by the "Potsdamer InnoLab für Arktisforschung" grant no. F221-08-AWI/001/002, namely the Brandenburg Ministry for Science, Research and Culture. I acknowledge the use of Perplexity (https://www.perplexity.ai) to assist with rewriting and improving the clarity and style of certain sections of this manuscript. The tool was used to rephrase and edit text based on my original drafts; all content

360  was critically reviewed and revised by the author prior to submission.





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
