# Peer review of "BorFIT: A Novel LiDAR-Based Training Dataset for Individual Tree Segmentation and Species Detection in northern boreal Forests"

_Earth System Science Data, 2025_

## Author Comment (AC1)

**General reply**

Dear Referee,

We thank you for the critical assessment of our dataset paper. BorFIT is intended primarily as a training dataset for artificial-intelligence (AI) applications. These applications will thereby improve our understanding of the boreal forest's vegetation reorganization. BorFIT itself already enables 3D spatial analysis of species distribution and stand structure across the circumboreal region. The reviewed preprint provides detailed information on how BorFIT was created, validated, and what the strengths and weaknesses of the dataset are. We agree with the comments regarding the overall clarity and structure of the manuscript and will integrate them when editing an updated version.

Best

Jacob Schladebach

Our response regarding specific comments:

**Technical description**

We agree with the suggestions to introduce more specificity in the descriptions of the technical setup used for data collection and the limitations of such acquired data:

- "It is very important to make clear here that the source of the data collection is UAV laser scanning (ULS). This gives the user insight into the type of spatial resolution in this dataset. Consider including this information also in the abstract."

- "Was any checkpoint or other feature used to verify the accuracy of the georeferencing? What is the estimated georeferencing accuracy?"

- "I think the specifications of the GNSS/INS system used and the expected accuracy of the position and attitude estimation for direct georeferencing are much more important here and should be clarified. Also, how the laser scanning system was set up is not clear: frequency used, angular resolution, FoV, angle of aperture, laser divergence, etc. The table with all the specifications of the sensors used is important to facilitate future users' information."

The manuscript will be amended with average values for the georeferencing accuracy, and we will clearly state the use of UAV-LiDAR. The technical details of the laser scanner are provided in Appendix Table A1.

**Comparison to other datasets and methodologies**

We especially agree with the need to expand the comparison of BorFIT with other datasets serving similar purposes, as suggested:

- "There are other datasets that provide individual trees, species, and leaf–wood separation, which could also be mentioned, even if they contain fewer samples. Extending the review here would improve the contribution discussion of this dataset and clarify the differences between the proposed open dataset and existing ones, particularly in terms of coverage, spatial resolution, and temporal resolution."

The introduction and discussion will be extended with more comparisons to highlight BorFIT's novelty. For example, ForINstance is currently mentioned only in the discussion but will be included in the Introduction to emphasize existing knowledge gaps we aim to close, including focus on circumboreal forests, spatial coverage, and quantity.

- "P1, Lines 27–30: I am not sure if I agree with the authors that this is the most used method nowadays. Some decades ago, yes, but nowadays there are many approaches, and stem detection has increased a lot, especially for boreal forests where the stems are often well visible. However, I agree that canopy-top or stem detection are the most common approaches for tree detection, although there are many other methods for segmenting trees depending on the spatial resolution, for instance, cluster-based ones are also very popular. I suggest mentioning more than one methodology here ('the most used are') and linking with some review."

The introduction will be expanded to include more recent methodologies for individual tree segmentation. However, the key message, that manual segmentation was more accurate in our case, remains.

**Selection of variables**

- "How were these eleven structural and two spectral variables selected? Could the authors link with previous works that also performed species classification to give more support to the method and chosen features?"

The spectral variables are based simply on what is possible with only RGB data. The structural parameters were chosen to simplify geometric shapes of crowns and point density distribution. The geometric shape fitting was inspired by the study by Qian et al. (2023), "Tree Species Classification Using Airborne LiDAR Data Based on Individual Tree Segmentation and Shape Fitting," as mentioned in the manuscript. More variables, including intensity, were tested but did not add meaningful value to the random forest performance, so they were excluded.

**Validation**

- "Regarding data validation, the species classification results could be discussed in more detail, clearly stating which classes users should be more cautious with when using the dataset. Including a table with estimated performance per species could be very helpful."

A table with species-specific classification errors is provided in the appendix (Tables A2–A5). When editing the updated manuscript version, it will be included in a separate validation section.

---

## Author Comment (AC2)

**General reply**

Dear Referee,

We thank you for the critical assessment of our dataset paper. BorFIT is intended primarily as a training dataset for artificial-intelligence (AI) applications. These applications will thereby improve our understanding of the boreal forest's vegetation reorganization. BorFIT itself already enables 3D spatial analysis of species distribution and stand structure across the circumboreal region. The reviewed preprint provides detailed information on how BorFIT was created, validated, and what the strengths and weaknesses of the dataset are. We agree with the comments regarding the overall clarity and structure of the manuscript and will integrate them when editing an updated version.

Best

Jacob Schladebach

Our response regarding specific comments:

**Data set and Pangaea**

*"Why the link to Pangea paper is included? I think only the reference is enough. Only the links to your dataset should be presented on the Data availability section to not create confusion. Actually, six links are presented which make a bit confusing how to actually find the BorFIT data set as a whole. Maybe the related five links could be referenced somewhere else and only on link to BorFIT data set should be clear presented on Data availability? Also would be nice to have some indication about the documentation related to the R code."*

A readme will be added to the code repository. The "data availability" section in its current state includes the raw data used for BorFIT. However, since they are mentioned in Table 1, they can indeed be excluded from the Data availability section.

*"When checking the Pangea repository and connecting here with the paper it is also good to include in the Data availability how the file is named and how the data was organized in the repository, for instance that they are presented in plot level. Also it is important to mention that the classes Trees and Species are extra bytes on Laz files and need to be considered when reading the files, since they are not standard fields on Laz structure. What is saving in the Classification and Point Source ID field? They look a bit strange. Please check."*

The Pangaea repository includes a product guide which provides the requested information. Classification is the scalar field that indicates the ground classification values, based on LASTools (1 = unclassified, 2 = ground). Point Source ID corresponds to groups of points acquired on specific flight lines.

*"Is EN23608(1)reference_plot_10_predicted.laz georeferencing correct? Looks that the trees are some how not vertical do the ground or the hill is very deep but the coordenates are somehow not aligned with E, N, up? Could the authors double check?"*

The cloud is indeed located at a steep slope. I can not see any issues here.

---

## Author Comment (AC3)

**General reply**

Dear Referee.

We thank you for the thoughtful assessment of our dataset paper. The boreal forest forms a continuous circumpolar biome, and vegetation reorganization processes transcend regional boundaries. Keeping the dataset unified acknowledges this continuity and supports research on large-scale drivers of change. At the same time, the metadata allows filtering by region, giving users the flexibility to create tailored subsets without fragmenting the dataset.

Nevertheless, we recognize that many users will focus on specific subsets, such as Yakutia in Eastern Siberia with its summergreen larch forests or the spruce-dominated stands of North America. To support this user-demand better as requested, we will update the PANGAEA metadata to include region, expedition, dominant tree species, and the random forest model used for prediction for each file, as we already discussed with the responsible PANGAEA editor. This will enable users to efficiently extract subsets relevant to their research questions.

We also agree with the comments on improving clarity in our descriptions and will address them in the revised manuscript. Find below answers to more critical comments.

Best regards

Jacob Schladebach

Our response regarding specific comments:

**Comment:**

I disagree that this is the most common approach for segmentation. There's plenty of research on satellite, airborne, and UAV lidar for segmentation, and even imagery-based approaches are more likely to use hyperspectral imagery than just color and IR.

**Response:**

We agree, that techniques based on true colour and infrared are outdated and other approaches are more common these days. We will update the text to include more recent techniques.

**Comment:**

I was confused by this segment. You appear to be attempting to describe why the boreal region presents special challenges, but none of these thoughts coalesce into a coherent explanation. Naming individual data sets in this context is similarly confusing. Consider rephrasing to make the challenges of this region more obvious. Suggest introducing BorFIT in a new paragraph.

**Response:**

The aim of the paragraph is to define the need for data sets, that close the gap between individual tree assessment and large-scale observations. We agree, that the specific challenges of the boreal region should be outlined separately and will edit the paragraph accordingly.

**Comment:**

Line 71: Please provide more detail about how sites were positioned. Were there formal criteria used or was this more ad-hoc?

Table 1: I have a lot of questions about how sites were selected. Why the large variation in number of point clouds and reference plots per region? Why was Alaska sampled twice?

**Response:**

The explanation for the site selection will be expanded. As mentioned in the manuscript, the primary positioning of field sites was based on satellite derived vegetation density (NDVI) and change detection, meant to cover different stand densities and fire scars. Further, very-high resolution imagery e.g. ESRI satellite imagery, was used to manually select transects in different forest types and structures. The transects in North America were established along highways and during different expeditions, that reach from the southern mixed forests towards the treeline and tundra. There is one Canadian transect established in 2022 covering especially the tundra and treeline ecotone in the north. Subsequently, the Alaskan transect established in 2023 covering forests and mountain treelines in the regions Southcentral, Interior and Far North, and in 2024 we went to western Alaska to the south of the Seward Pensinsula to investigate the westernmost forests and treeline ecotone. During the last expedition included in this dataset (Alaska 2024), we were additionally revisiting formerly investigate sites in Alaska Interior and were additionally establishing new ones in this region. This is the reason why Alaska was sampled twice, at this point. However, BorFIT includes data at each reference plot from on point of time only, and data from Alaska 2024 were not examined the year before.